# DEEP CONVOLUTIONAL MALWARE CLASSIFIERS CAN LEARN FROM RAW EXECUTABLES AND LABELS ONLY

**Marek Krčál**
Czech Academy of Sciences
krcal@cs.cas.cz

**Ondřej Švec**,[*] **Otakar Jašek**
Avast Interns
{ond.svec,jasek.ota}@gmail.com

**Martin Bálek**
Avast
balek@avast.com

## ABSTRACT

We propose and evaluate a simple convolutional deep neural network architecture detecting malicious *Portable Executables* (Windows executable files) by learning from their raw sequences of bytes and labels only, that is, without any domain-specific feature extraction nor preprocessing. On a dataset of 20 million *unpacked* half megabyte Portable Executables, such end-to-end approach achieves performance almost on par with the traditional machine learning pipeline based on hand-crafted features of Avast.

## 1    INTRODUCTION

One of the cornerstones of modern deep networks is the approach of *end-to-end learning*, or equivalently, *automatic feature extraction* where only the labels and raw data are presented to the network with no hand-crafted features provided and close to no preprocessing.

The end-to-end approach has not yet gained the dominance in the field of malware detection, which is a field of growing importance and market value driven by ever growing malicious software (*malware*) development.[1] Despite several interesting results we mention below, we are aware of no published attempts to train end-to-end neural network classifiers on an *industrial-sized dataset* of clean and malicious files. In this paper, we present simple but successful convolutional networks trained and evaluated on 20 million Windows executable files (so-called *Portable Executables*) represented as plain sequences of bytes. This work explores the limits to which a standard (baseline) architecture can get: embedding layer followed by four convolutions with strides and a max pooling in the middle, global average pooling and four fully connected layers. See Section 2 for details and the additional discussion of our design choices that tuned the architecture for the regime of very small false positive rates. Our approach is the end-to-end counterpart of so-called *static malware analysis*: the network is given mere sequence of bytes that the executable consists of. Nonetheless, we expect that similar architectures would give good results on the end-to-end variant of the dynamic malware analysis, where the network would be given the machine code or other low level representation to which an *emulator* or a *sandbox* unwraps the portable executable. A result in that direction by Huang and Stokes (2016) combines some general feature extraction from a sandboxed emulation and intensive representation learning.

**Portable Executables—unconventional data type for deep learning.** The Portable Executable is a complex format with only local sequentiality (1-D structure) and with the meaning of its byte symbols very diverse and context dependent—e.g., in the context of header, in sections of various types, resources or relocation tables. Hence it is natural to ask to which extent the well established deep learning architectures can learn from such raw input. In addition, we have chosen the Portable Executables as they are by far the most severe channel for security threats on a PC. We believe that large scale experiments on such a relevant and under-investigated data domain could be of interest to the whole community of deep learning research.

**The dataset.** Out of the Avast's repository of PE files we have chosen all those collected during recent 16 months of size between 12 and 512 kilobytes excluding files with some obfuscation methods

---

[*]Currently in Google.
[1]Business Insider estimated that $386 billion will be spent on securing PCs—see Insider (2016)—and malware detection is a major segment in there.

such as compression or encryption detected.[2] The train, validation and test sets consists of the first 12 months, the next 2 months and the last 2 months, respectively, so that we measure how the model generalizes into the future. For the sake of simplicity, we use binary labels *clean* and *malware* only with roughly balanced occurrence throughout our dataset.

A great obstacle hindering the public research on learning malware classifiers (and deep network classifiers in particular) is the lack of an industrial-sized publicly available datasets. This causes fragmentation of the research where different results are mutually directly incomparable, if reproducible at all. It is our longer-term vision to make available some of our data in a form and volume that would be appealing to the deep learning community.

## 2 ARCHITECTURES

The scheme of our network is visualized in Figure 1; several remarks follow.

**Fixed embedding.** Each byte of the input sequence is first embedded to an 8-dimensional vector of the form $(\pm 1/16, \ldots, \pm 1/16)$ according to its binary representation where constant $1/16$ was found empirically. We observed no performance difference between learnable and non-learnable embeddings.

**Convolutions with stride—reducing the computational load.** To mitigate the computational burden, we apply experimentally tuned strides of 4 and 8 at the first and the second block of convolutions, respectively. We have verified that using strides of 3,5,7 and 9 (non-powers of two) in the respective order causes relative drop roughly by 6–10 percents in all the metrics we have measured.[3]

**Details on training.** We initialize the convolutional layers by random values drawn from the uniform distribution according to Glorot and Bengio (2010) and the fully connected layers according to Klambauer et al. (2017). The training loss is the usual cross-entropy with every clean sample contributing to the loss 7 times as much as every malicious sample. We group the executables into batches of 128 similarly sized files padded by zeros at the end (right padding). The network is trained by the Adam optimizer (Kingma and Ba (2014)) with the default parameters. According to the scores on the validation set, we stop the training shortly after the third epoch.

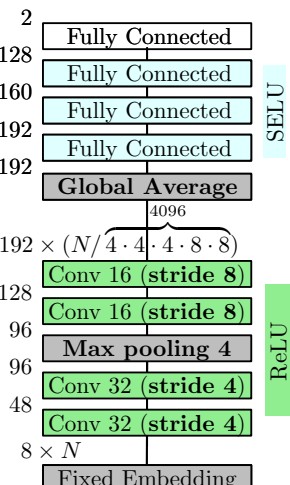

Figure 1: Our convnet.

**Choices specifically driven by the zero false positives target.** Malware detectors are tuned for low false positive rates so that they do not overwhelm users by false malware detections under the real distribution with the vast dominance of clean files. We formalize this target score as the area under the Receiver Operator Curve restricted to the interval $[0, 0.001]$ of the false positive rate. For convenience the area is reported in percentages of the maximal possible such area—0.001. We will refer to such score as the restricted AUC. Below we list possible changes in our architecture and the corresponding estimated drops in the restricted AUC score. On the other hand, each of the variations improves cross-entropy and/or accuracy:

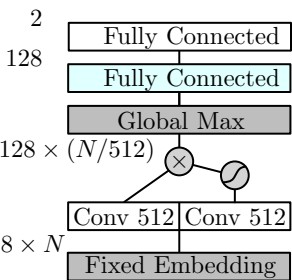

Figure 2: MalConv 1.1. Both kernel size and stride is 512.

1. Global Max instead of Global Average: -20% relative drop.

2. Clean and malware files with equal weight: -10% relative drop.

---

[2]Our pipeline could be extended to cover majority of the obfuscated files by using *unpackers*.

[3]Strides and pooling lengths is the only hyper parameter of our network consciously tailored to the executables: all compilers (e.g., Microsoft Visual C++) align the beginnings of so-called sections within the executable to multiples of powers of two (e.g., 4096).

Table 1: Reported scores based on 10 (the third row) or 4 independent runs (the rest)

| classifier | restricted AUC | cross-entropy | accuracy |
|---|---|---|---|
| **MalConv** | $66.1\% \pm 0.9$ | $0.204 \pm 0.028$ | $94.6\% \pm 0.6$ |
| **Our convolutional network** | $70.4\% \pm 0.5$ | $0.165 \pm 0.020$ | $96.0\% \pm 0.6$ |
| **FNN on handcrafted features** | $73.2\% \pm 2.3$ | $0.151 \pm 0.015$ | $96.2\% \pm 0.3$ |
| **FNN on enriched features** | $76.1\% \pm 1.0$ | $0.114 \pm 0.006$ | $97.1\% \pm 0.2$ |

3. ReLU instead of SELU in the fully connected layers: -4% relative drop.

**Related work—baseline.** Raff et al. (2017) have recently developed a convolutional architecture called MalConv for exactly the same task but using a dataset order-of-magnitude smaller than ours. The presented architecture has been developed completely independently and we found no improvement by employing elements from MalConv. On the contrary, we have slightly increased the performance of MalConv on our dataset by using power-of-two strides, SELU activation and removing the DeCov regularization, see Figure 2. Nonetheless, the MalConv's performance is very good given the limited dataset used for the development, see Table 1.

## 3 EVALUATION

**Learned versus handcrafted representations.** We compared the learned convolutional features against a collection of in-house 538 static features from a machine learning system aimed at rapid feature prototyping by the malware analysts at Avast. To this end we trained and evaluated feedforward network (FNN) of the shape 538–512–256–192–168–128–2 with SELU activation and wight decay of $3 \cdot 10^{-7}$ on the same collection of files. Finally, to measure how much the learned representation is complementary to the hand-engineered one, we ran a slightly wider feedforward network on the enriched feature vector obtained by adding the 192 features output by the average operation of our convnet. For the results see Table 1.

The best results of the ensemble approach demonstrate that learning from raw executables serves as a valuable new feature engineering process. Remarkably, learning jointly from raw executables and the hand-engineered features might give even better results.

**Localization with grad-CAM.** We use the gradient-weighted class activation mapping (Grad-CAM) (Selvaraju et al. (2017)) to find the blocks within files that contribute the most to the malware prediction of our convolutional network. We let the malware analysts to judge the relevance of the highlighted blocks on several selected executables. Not always, but very often they found something suspicious there: header of a Portable Executable embedded within a Portable Executable or a "VERSION_INFO" segment with a fake vendor and software name in a Xindl virus or list of very unusual imported API functions in the case of the ransomware Locky.

**Conclusion.** On the one hand, the end-to-end learning from raw executables and labels only is still slightly behind the reference ML pipeline even on a dataset chosen favourably for the convolutional networks. On the other hand, judging from the long-lasting improvement rate of deep learning models in the fields like computer vision, machine translation or speech recognition, we forecast significant performance gains by new architectures and also by learning from larger and more carefully crafted datasets. There is an orthogonal production-aimed direction of inserting some domain knowledge while leaving room for intensive representation learning: using malware families as refined labels (Huang and Stokes (2016)), enriching the byte sequence by other signals such as entropy rate (Saxe and Berlin (2015)) or tailoring the architecture more specifically to the format of Portable Executables. A speedup of training would be an important catalyst for further improvements and production applicability: here innovations like depth-wise separable convolutions by Chollet (2017) or sparsely gated mixture of experts by Shazeer et al. (2017) might help. Last but not least, we believe that deep learning can eventually benefit its human teachers by observing and localizing previously unseen patterns in malware and clean files.

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
