# OpenReview forum: "Deep Convolutional Malware Classifiers Can Learn from Raw Executables and Labels Only"
_ICLR.cc/2018/Workshop — Accept_

### Official Review · AnonReviewer2 · 2018-02-26
**Interesting idea, slightly negative result, but solid work**

**Rating:** 6
**Confidence:** 3

**Review:**

The paper presents an end to end approach for malware detection based on byte sequences.

+ The authors added strong baselines with hand crafted features
+ the authors attempted to understand the decisions of the network

- files with obfuscation methods were excluded
- The approach does not perform as well as hand engineered features
- not all details given

Interesting idea and in my opinion good enough for the workshop.

---

### Official Review · AnonReviewer1 · 2018-03-08
**interesting application of CNNs**

**Rating:** 6
**Confidence:** 3

**Review:**

This paper proposes to use a convolutional neural network, taking program binaries as inputs, to detect malware. The results are on par with a network that takes in hand-crafted features, and the hidden layers are shown to be complementary to the hand-crafted features.

The paper is well written and easy to follow. The descriptions of the experiments are complete. One minor thing missing is the learning rate of the Adam optimizer.

I have a more fundamental question about the task. How much does the model learn to identify parts of the problematic source code? How much does the model learn to identify signs unrelated to the source code, such as things in the header? I assume that in many cases you can judge whether a piece of binary is malware just by looking at the cosmetics of the program, but these features might not generalize well in the long run. I assume it would be much more useful to identify certain sequence of instructions and memory addresses based on cpu architectures. It would be great to analyze the CNN to see what it is paying attention to.

The other fundamental question about the task is obfuscation. On the one hand, in theory there is no universal tool to overcome obfuscation (with the standard crypto assumptions). On the other hand, it's probably easy to detect trivial obfuscation, because the resulting program binaries would look significantly different from regular program binaries. How often are binaries obfuscated in general? How do these affect the model's performance? What are the characteristics of regular program binaries?

Programs as data are more stringent than natural images and words. The CNNs might require a lot of effort just learning the common idioms of programming, such as control flows, loops, from the binaries. I think the CNNs might benefit from these high-level constructs/templates when identifying malwares. Since CNNs or deep models in general are strong at coping with noise, it might be useful to look at how they behave on mutated malware.

---

> ### Author Response · Authors · 2018-04-06
> **answers**
>
> 1) According to the latest experiments with Guided Backpropagation combined with Grad-CAM, we saw no "explanation" based on a machine code (however, the number of manually inspected samples and the time spent have been very limited so far). In this aspect the automatic convolutional features might be very similar to the traditional hand-engineered features which cover mainly the "envelope" of the executable
>
> 2) We do think that obfuscation can be overcome in theory, emulation/sandboxing being one (relatively) universal tool. However we do not expect the convnet to do the job. Because of the type "explanations" we have observed so far, we think that the performance may not be hurt significantly by including obfuscated files (indeed we use fairly simple/imperfect detection of obfuscation and our dataset certainly contains a large portion of obfuscated files already). Tests on the whole collection will be hopefully ready soon.
> Obfuscation is quite common for the current malware but majority can be reversed by manually crafted "unpackers", that is another option.
>
> 3) We believe that we can force the deep nets to focus on the code itself is by inputting emulation/sandboxing output instead of the file. That is definitely one of the major directions for further research.

---

### Official Review · AnonReviewer3 · 2018-03-10
**A new convolutional network architecture for malware classification**

**Rating:** 7
**Confidence:** 4

**Review:**

The new network architecture proposed in the paper is novel. The experiment results show this architecture compares favorable with MalConv on a data set collected by the authors. It is good to know that learned features also help to improve accuracy. Several questions:
1) Are the malicious and benign examples balanced?
2) Is the good performance due to its deeper structure comparing with MalConv? How long is training time?
3) Why does SELU help here?

---

> ### Author Response · Authors · 2018-04-06
> **answers**
>
> 1) Yes, clean and malicious examples are roughly balanced in the dataset
> 2) We have not experimented extensively with the depth but we were not able to improve upon the shallow MalConv, so we assume that depth helps. Training time is between 2-3 days on the full dataset.
> 3) After recent repeated multiple experiments with sole ReLU activations in all layers, we found no measurable difference. We apologize for a hurried conclusion in our abstract based upon a single run (each run is simply too expensive for us).

---

### Author Response · Authors · 2018-03-12
**Training fails with batch normalization**

While compressing the paper into the 3 page abstract, we have accidentally omitted a relatively important remark: Adding batch norm to our architecture yields no speedup and severely damages the model's performance. Raff et al. report the same observation in the case of their MalConv architecture and its variants.

---

### Decision · Program_Chairs · 2018-03-20
**ICLR 2018 Workshop Acceptance Decision**

**Decision:**

Accept

**Comment:**

Congratulations, your paper was accepted to the ICLR workshop.